# Electrodeposition of Pd-Pt Nanocomposites on Porous GaN for Electrochemical Nitrite Sensing

**DOI:** 10.3390/s19030606

**Published:** 2019-01-31

**Authors:** Rui Xi, Shao-Hui Zhang, Long Zhang, Chao Wang, Lu-Jia Wang, Jing-Hui Yan, Ge-Bo Pan

**Affiliations:** 1College of Chemistry and Environmental Engineering, Changchun University of Science and Technology, Changchun 130022, China; rxi2017@sinano.ac.cn; 2Suzhou Institute of Nano-tech and Nano-bionics, Chinese Academy of Sciences, Suzhou 215123, China; shzhang2016@sinano.ac.cn (S.-H.Z.); lzhang2017@sinano.ac.cn (L.Z.); chaowang2017@sinano.ac.cn (C.W.); ljwang2016@sinano.ac.cn (L.-J.W.)

**Keywords:** porous GaN, electrodeposition, Pd-Pt nanocomposites, nitrite detection

## Abstract

In recent years, nitrite pollution has become a subject of great concern for human lives, involving a number of fields, such as environment, food industry and biological process. However, the effective detection of nitrite is an instant demand as well as an unprecedented challenge. Here, a novel nitrite sensor was fabricated by electrochemical deposition of palladium and platinum (Pd-Pt) nanocomposites on porous gallium nitride (PGaN). The obtained Pd-Pt/PGaN sensor provides abundant electrocatalytic sites, endowing it with excellent performances for nitrite detection. The sensor also shows a low detection limit of 0.95 µM, superior linear ampere response and high sensitivity (150 µA/mM for 1 to 300 µM and 73 µA/mM for 300 to 3000 µM) for nitrite. In addition, the Pd-Pt/PGaN sensor was applied and evaluated in the determination of nitrite from the real environmental samples. The experimental results demonstrate that the sensor has good reproducibility and long-term stability. It provides a practical way for rapidly and effectively monitoring nitrite content in the practical application.

## 1. Introduction

As a typical inorganic pollutant, nitrite is closely related to pluralistic fields, such as environmental chemistry, food industry and biological process. The contamination nitrite in ground water mainly comes from agricultural fertilizer and industrial effluent, which can infiltrate the soil [1,2,3]. Meanwhile, nitrite is relevant to the formation of carcinogenic nitrosamines by reacting with secondary amines, and it can bind to hemoglobin to cause methaemoglobinaemia in infants [4,5]. Therefore, the accurate and on-site determination of nitrite is very important for both environmental protection and live processes [6]. At present, various techniques are used to detect nitrite, such as fluorescence spectrometry [7], chromatography [8], chemiluminescence [9] and electrochemical methods [10]. Among them, electrochemical methods have overcome the shortcomings of other techniques, such as expensive instruments, complex operations and complicated sample preparations, which have attracted the wide attention of researchers due to their low cost, speed, good sensitivity and selectivity [11,12].

The purpose of chemically modified electrodes is to carry out the sensing material design on the electrode surface. Up to now, much research has been reported on electrode modified materials, such as carbon material graphene [13], metal-organic frameworks (MOFs) [14] and enzyme [15]. Carbon materials have good conductivity and can be commonly used as a substrate for supporting electro-active substances. Metal-organic frameworks (MOFs) possess high surface area and porosity characteristics. Enzyme shows high specificity and the efficient catalysis. However, some problems on the above modified electrodes are still not well solved in practice. Firstly, most modified materials are mainly adsorbed to the surface of the electrode by physical adsorption. In the long-term use, modified materials will peel off due to scratches and mechanical actions, affecting the electrode catalytic activity and lifetime. Secondly, modified electrodes are not suitable for harsh solution environments. For instance, graphene can easily be decomposed in strong acidic solutions. The enzyme activity can be easily affected by the temperature and the pH value of the solution; it is hard to guarantee the stabilization of the electrode.

The properties and structures of sensing electrodes are therefore particularly important. However, gallium nitride (GaN), a representative wide bandgap semiconductor, could be a promising electrode candidate owing to its large potential window, high electron mobility and strong metal-semiconductor interaction for electrochemical sensors [16,17,18]. Compared with other electrode materials, GaN can achieve a higher chemical and thermal stability. These unique properties make it generate less noise, resulting in a low background signal. Moreover, porous GaN (PGaN) with different pore sizes and shapes has been easily obtained by wet etching. Compared with planar GaN, PGaN provides higher surface area and more defects. The correlation research indicated that it can be applied in some fields as an excellent electrode [19,20,21].

Due to their excellent electrocatalytic activities, noble metal nanomaterials (NMs) have been applied to composite electrochemical sensors [22,23,24,25,26]. In comparison with other nanomaterials, NMs are considerably more stable under ambient conditions. Moreover, their unique catalytic performances together with their decreased dimensions and high surface-to-volume ratio demonstrate enormous potentials in environmental and biological applications. In this work, a facile two-step electrodeposition process was developed to decorate the PGaN electrode with Pd and Pt (Pd-Pt) nanocomposites in order to fabricate a novel nitrite sensor. In the first step, Pt was electrodeposited on PGaN by cyclic voltammetry (CV). In the second step, Pd was electrodeposited on the as-prepared Pt-modified PGaN (Pt/PGaN) by chronoamperometry (CA). The obtained Pd-Pt/PGaN sensor provided a wide linear range, lower detection limit, high selectivity and excellent stability for nitrite. Moreover, the sensor exhibited a good recovery for nitrite determination in practical sample analysis. 

## 2. Experimental Section

Chemicals: Chloroplatinic acid (H_2_PtCl_6_), palladium chloride (PdCl_2_), sodium nitrite (NaNO_2_), disodium hydrogen phosphate (Na_2_HPO_4_), potassium dihydrogen phosphate (KH_2_PO_4_) and some other chemicals were analytical reagents, purchased from Sinopharm Chemical Reagent Co., Ltd. (Shanghai, China). 1-Ethyl-3-methylimidazolium trifluoromethanesulfonate ([EMIM][OTF]) was provided by Shanghai Cheng Jie Chemical Co., Ltd. (Shanghai, China). Phosphate buffered saline (PBS) (0.1 M pH 7.0) was prepared by KH_2_PO_4_ and Na_2_HPO_4_ according to a certain volume ratio. A fresh solution of NaNO_2_ was prepared daily.

Characterization: All the electrochemical experiments were accomplished by Autolab workstation (Metrohm PGSTAT302N, Metrohm AG, Herisau, Switzerland). The surface morphology of all samples was examined by Hitachi-S4800 scanning electron microscope (SEM, Hitachi (Hong Kong) Ltd. Hong Kong, China). The X-ray diffraction (XRD) pattern was recorded by Bruker D8 Advance power X-ray diffractometer (Bruker, Billerica, MA, USA). The elemental analysis was obtained by energy dispersive spectroscopy (EDS, Quanta FEG 250, Thermo Fisher Scientific, Waltham, MA, USA). 

Fabrication of PGaN: A 5 μm thick single-crystalline GaN layer with the Si-doped was grown on (0001) c-plane sapphire substrates by hydride vapor phase epitaxy (HVPE). The size of GaN chips was 0.3 cm × 1.5 cm. Photoassisted electrochemical etching (PECE) was completed in a two-electrode system. The etchant was 1-Ethyl-3-methylimidazolium trifluoromethanesulfonate. 

In the typical etching process, the front-side vertical illumination source was supplied by a 300 W Xenon lamp. This work has been reported about in relation to our group [27]. The etching voltage applied to the electrode system was 7 V for 15 min. The as-obtained PGaN electrode was ultrasonically cleaned in acetone, ethanol and deionized (DI) water for 20 min to remove any surface substance. Then, the PGaN electrode was dried with an N_2_ stream.

Synthesis of Pd-Pt/PGaN: The synthesis of electrodes was divided into two steps. Firstly, Pt was prepared by CV in a classical three-electrode system using PGaN as a working electrode, a Pt plate as a counter electrode, and Ag/AgCl as a reference electrode. The electrolyte solution comprised 4 mM H_2_PtCl_6_ and 0.5 M NaCl. The Pt/PGaN electrode was synthesized by optimizing the deposition conditions. The range of voltage was from −1.5 V to 0.5 V, the scan rate was 50 mV/s and the number of cycles was four. The as-obtained Pt/PGaN electrode was thoroughly rinsed by DI water and used for the next experiment. Secondly, Pd was deposited by CA using Pt/PGaN as the working electrode. The electrolyte was replaced with 5 mM PdCl_2_ and 0.5 M NaCl. Thirdly, the prepared Pd-Pt/PGaN electrode was thoroughly washed with DI water and dried with an N_2_ stream.

Electrochemical nitrite detection: Nitrite detection was studied in a conventional three-electrode system using a Pd-Pt/PGaN sensor. CV curves were used to record the anodic peak current response to 1 mM nitrite in 0.1 M PBS. The CA measurement was successfully achieved by adding different concentrations of nitrite into the PBS solution with a constant stirring rate of 350 rpm. All the electrochemical experiments were performed at room temperature.

## 3. Results and Discussion

Figure 1a shows the SEM image of a PGaN electrode fabricated by PECE. One can see that the pore shape is nearly hexagonal, which is in accordance with the hexagonal system structure of GaN [28]. The PGaN electrode has a high density pore structure and homogeneous pore distribution. Figure 1b shows that the pore diameter of PGaN is approximately 40-130 nm and that the average pore size is 90 nm. The whole etching process was accomplished under the UV-light Xenon lamp vertical irradiation, and the etchant was a nonaqueous ionic liquid. As described in our previous work, the formation and dissolution of Ga(CF_3_SO_3_)_3_ during the interface between GaN and the ionic liquid contributes to the formation of porous structures [20]. The PGaN with a high-porosity structure has the higher specific surface area than planar GaN, and it can effectively increase the loading amount of the catalyst.

Figure 2a shows the SEM image of a Pt/PGaN electrode. The Pt nanoparticles were uniformly distributed throughout the pore structure of PGaN. Figure 2b shows the morphological features of Pd nanoparticles fabricated by CA method on a PGaN electrode. All of their forms were inhomogenous in density and size, with diameters from 10 nm to 80 nm. The Pd/PGaN electrode was only used for comparative studying. Figure 2c shows the SEM image of a Pd-Pt/GaN electrode. As can be seen, the Pd-Pt nanocomposites were well-distributed on the whole PGaN electrode surface. By selecting and optimizing the experimental conditions, nanocomposites can be prepared by a simple two-step electrodeposition process. In the first step, the preparation of the Pt catalyst supported on a PGaN electrode by CV was conducted in a chloroplatinic acid electrolyte. In the case of high deposition overpotential, the deposition process of metal particles follows the instantaneous nucleation mechanism [19]. The negative potential of CV was therefore set to −1.5 V, which was beneficial to the growth of small-sized Pt particles. In the second step, Pd nanoparticles were not only deposited on the pore walls of the PGaN, but also in combination with metal Pt to form the nanocomposites. A constant potential electrodeposition was carried out at −1.5 V, and a uniform, compacted and good adhesive active layer was obtained. During the whole process, the pore structure of PGaN can effectively restrain the agglomeration of nanoparticles.

Figure 3a shows the EDS analysis of a Pd-Pt/PGaN electrode. The peaks of Pd and Pt were clearly observed in the spectrum. Apart from the Ga and N, the peaks of no other elements were present. The crystal planes of Pd-Pt nanocomposites were measured by XRD, as shown in Figure 3b. The diffraction peaks positioned at 39.91°, 46.39° and 67.96° correspond to the (111), (200) and (220) planes respectively. Compared with the diffraction peak of the Pt (111) crystal face at 39.7°, the Pd-Pt nanocomposite peaks are shifted to a higher angle due to the combination of Pd to Pt. The results of the experiment correspond to literature reports [29,30]. No characteristic peaks of Pd or Pt oxides were detected. Therefore, the nanocomposites composed of Pt and Pd were successfully deposited on the PGaN via the electrochemical method.

Figure 4a shows the electrocatalytic responses of the Pd-Pt/PGaN and PGaN electrodes to 1 mM NaNO_2_ at the potential range from 0.2 to 1.2 V. Compared with the CV curve a_2_, a conspicuous oxidation peak of the Pd-Pt/PGaN electrode was observed at about 100 µA at 0.65 V in curve a_1_. The result demonstrates that Pd-Pt nanocomposites play an important role in the electrochemical detection of nitrite. The significant change of the Pd-Pt/PGaN electrode may be due to the nanocomposites, based on the strong electronic and synergistic effect [31,32]. Meanwhile, no distinct current response was observed for the PGaN (curve b_1_ and b_2_). That is to say that the bare PGaN electrode provides a low background signal, which is beneficial for sensors to generate less noise and recognize weak signals. To further illustrate the excellent catalytic performance of Pd-Pt/PGaN, the effects of Pt/PGaN and Pd/PGaN on direct oxidation of nitrite ion were studied by CV. As shown in Figure 4b, the Pt/PGaN electrode demonstrated a small current response to 1 mM of nitrite, while no significant current change was observed on the Pd/PGaN electrode. The results show that Pd-Pt bimetallic complex has a higher catalytic activity than Pt single metal in the detection of nitrite. 

Figure 4c shows the Nyquist plot of three electrodes for electrochemical impedance spectroscopy (EIS) measurements in 0.1 M KCl containing 5 mM of redox couple [Fe(CN)_6_^3−^/^4−^] at 220 mV. The inset is the Randles equivalent circuit for fitting the experimental data. Rs represents the ohmic resistance of the electrolyte, Rct and Zw are the charge transfer resistance and Warburg impedance, respectively. CEP is the double layer capacitance, and the frequency is from 10^−1^ to 10^5^ Hz. The semicircle segment observed at higher frequencies conforms to the Rct, and a linear region that appeared at lower frequencies is attributable to the diffusion process of the redox couple. All the electrodes show different diameters of semicircles, exhibiting different Rcts. The Rct values of the Pt/PGaN and Pd/PGaN were approximately 1200 and 1700 Ω, respectively. After the addition of Pd, a low electron-transfer resistance to 120 Ω was obtained on the Pd-Pt/PGaN electrode. This change reveals that the nanocomposites, formed by the in-situ growth of Pd on Pt, accelerate the redox reaction. Additionally, the increase of the surface active region also leads to the decrease of Rct. In addition, the effects of the Pd-Pt/PGaN electrode with different pH conditions on the detection of nitrite were studied in Figure 4d. The results show that the peak current values of the electrode increase as the pH of the solution increases in the range of 2.0 to 7.0, and then begin to fall rapidly. Thus, 7.0 was selected as the optimum pH condition.

Figure 5a shows the peak current responses of a Pd-Pt/PGaN electrode to NaNO_2_ in a 0.1 M PBS buffer at an applied potential of 0.65 V under stirring. The Pd-Pt/PGaN electrode exhibited a timely increase response towards every addition of NaNO_2_, and the amperometric response attained a steady state in less than 3 s. Furthermore, the corresponding calibration plots of the response to the nitrite concentration are shown in Figure 5b,c. The current increases linearly with the concentration of NaNO_2_ in two ranges. One is from 1 to 300 µM with a linear regression equation of I (µA) = 0.903 + 0.150c (µM) and the correlation coefficient of 0.997. The sensitivity of the electrode is as high as 150 µA/mM. The other linear regression equation is the concentration range of I (µA) = 28.436 + 0.073c (µM) from 300 to 3000 µM. The sensitivity and correlation coefficients are 73 µA/mM and 0.996, respectively. The limit of detection (LOD) of this sensor is 0.95 µM with a signal-to-noise ratio of 3. The obtained results show that the Pd-Pt/PGaN electrode possesses excellent electrocatalytic activity towards the oxidation of nitrite.

The interference of a common substance in nitrite detection for the modified electrode was also examined and the result displayed in Figure 5d. When 5 mM UA, AA, DA, GLU, Na_2_CO_3_, KNO_3_, GaCl_2_ and MgSO_4_ were added into PBS solution respectively, no amperometric responses were detected. Nevertheless, a significant current response to 0.1 mM NaNO_2_ was timely, even in the presence of a large amount of interfering substance. That shows that the obtained Pd-Pt/PGaN sensor has a good selectivity for nitrite.

The reproducibility and stability of a Pd-Pt/PGaN electrode were tested in 0.1 M PBS containing 1 mM NaNO_2_, as shown in Figure 6a,b. The reproducibility was investigated by comparing the amperometric responses produced by different electrodes under the same condition of preparation. The peak current values of eight Pd-Pt/PGaN electrodes were 100.8, 103.4, 103.7, 101.6, 102.3, 100.9, 103.5 and 101.5 µA, with an RSD of 2.37%, as shown in Figure 6a. Moreover, the Pd-Pt/PGaN sensor was stored under atmosphere condition for 15 days to evaluate its stability, as shown in Figure 6b. The electrode retained 93% of the initial current response after 15 days. The above sensing performances confirm that the Pd-Pt/PGaN sensor exhibits both high reproducibility and stability in the nitrite detection process.

To estimate the sensing properties of Pd-Pt/PGaN sensors, the electrochemical analysis parameters of different electrode substrates were compared, as listed in Table 1. Although the detection limit of Pd-Pt/PGaN sensors is lower than some sensors, the linear range and sensitivity are wider and higher than most other Pt or Pd based sensors. The previously reported sensors were fabricated by employing complex procedures. On the contrary, the Pd-Pt/PGaN sensor was modified by a simple electrodeposition method. The good sensitive performance of the sensor is mainly attributed to the effective combination of PGaN and Pd-Pt nanoparticles. On the basis of its high electron mobility and special surface structure, GaN accelerates the electron transfer rate between the nitrite and electrode, which is beneficial to the proceeding of reaction. Therefore, the above results indicate that GaN is expected to become an excellent electrode for sensors.

The practical application of a Pd-Pt/PGaN electrode was examined by adding the known concentration of nitrite in tap and lake water samples, and the results were shown in Table 2. These experiments were carried out five times under the same conditions. Good recoveries ranging from 96 % to 106 % demonstrate that the sensor has high chances of being effective in a real environment.

## 4. Conclusion

We exhibited a novel Pd-Pt nanocomposites-modified PGaN electrode through a simple two-step electrochemical deposition route for nitrite sensing. For the high-porosity structure that uses PGaN as a supporting electrode and for the effective electronic transmission of Pd-Pt nanocomposites, the Pd-Pt/PGaN nitrite sensor presented many features, such as a wide linear range, high sensitivity, good selectivity and stability. Furthermore, it can detect the nitrite in tap and lake water, respectively. The simply assembled Pd-Pt/PGaN sensor provides a fast and effective method for monitoring nitrite in a realistic environment.

## Figures and Tables

**Figure 1 sensors-19-00606-f001:**
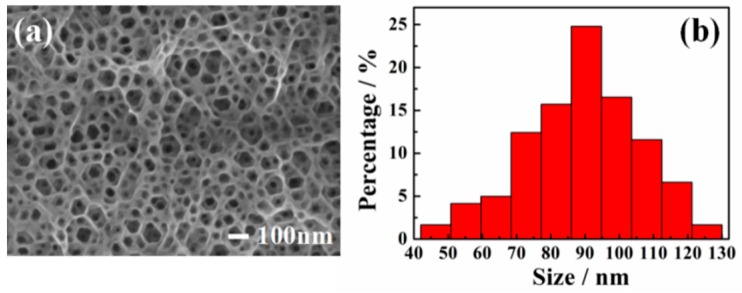
(**a**) SEM image and (**b**) aperture distribution histogram of PGaN electrode.

**Figure 2 sensors-19-00606-f002:**
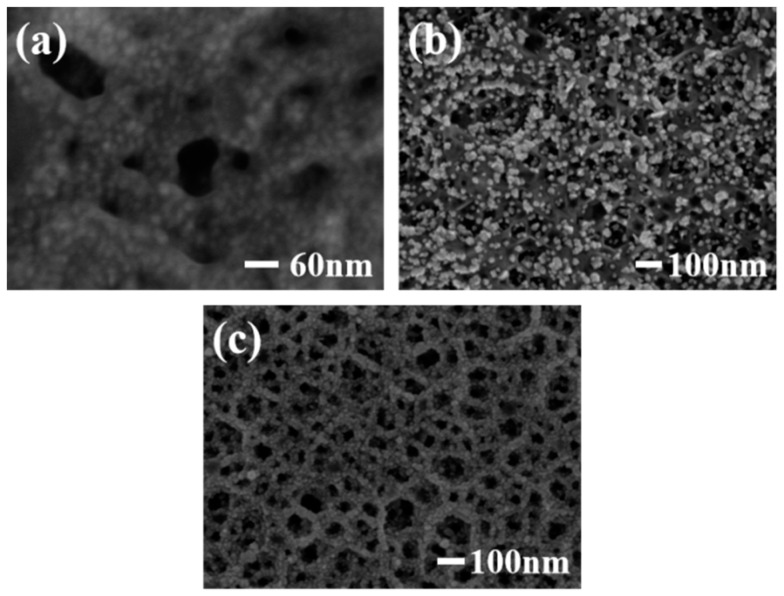
SEM images of (**a**) Pt/PGaN, (**b**) Pd/PGaN and (**c**) Pd-Pt/PGaN electrodes.

**Figure 3 sensors-19-00606-f003:**
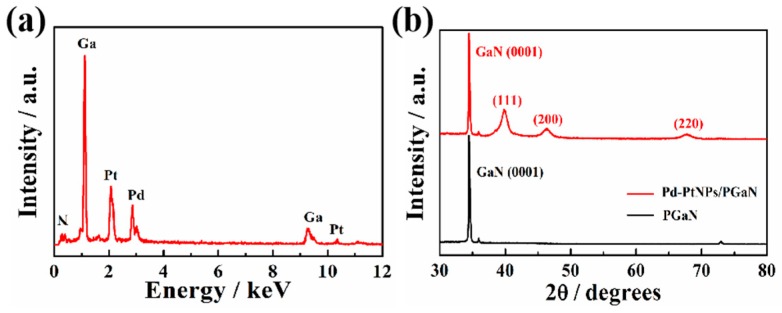
(**a**) EDS spectrum of the Pd-Pt/PGaN electrode. (**b**) XRD patterns of the PGaN and Pd-Pt/PGaN electrodes.

**Figure 4 sensors-19-00606-f004:**
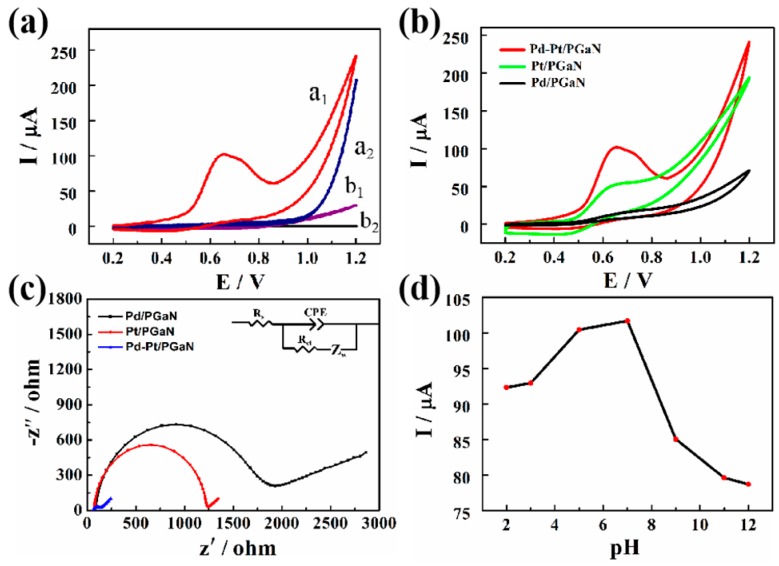
(**a**) CVs of Pd-Pt/PGaN electrode (a_1_ and a_2_) and PGaN electrode (b_1_ and b_2_) in 0.1 M PBS (pH 7) solution in the presence (a_1_ and b_1_) and absence (a_2_ and b_2_) containing 1 mM NaNO_2_. (**b**) CVs of the Pt/PGaN, Pd/PGaN and Pd-Pt/PGaN electrodes in 0.1 M PBS buffer containing 1 mM NaNO_2_. (**c**) Electrochemical impedance spectroscopy (EIS) spectra of three electrodes in 0.1 M KCl containing 5 mM K_3_[Fe(CN)_6_]/K_4_[Fe(CN)_6_]. The inset is the Randles equivalence circuit model. (**d**) CV peak current values of the Pd-Pt/PGaN electrode in 0.1 M PBS buffer containing 1 mM NaNO_2_ under different pH conditions (2.0, 3.0, 5.0, 7.0, 9.0, 11.0 and 12.0).

**Figure 5 sensors-19-00606-f005:**
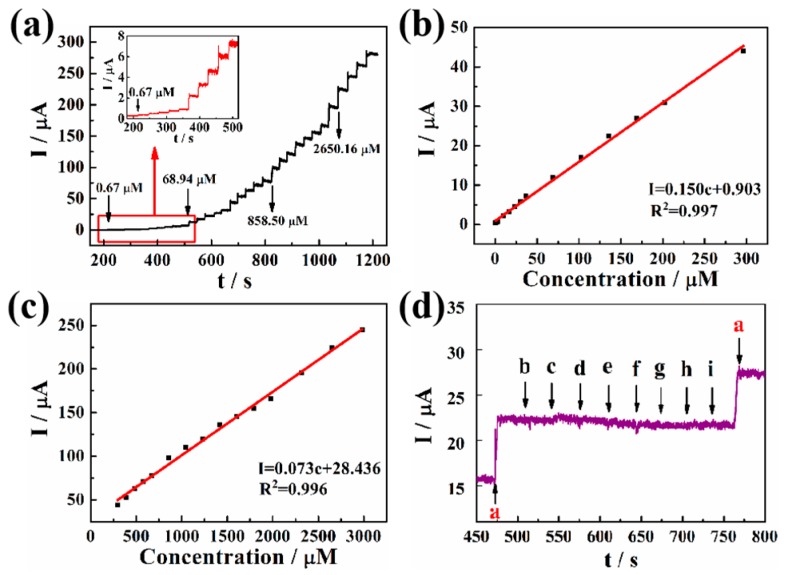
(**a**) Amperometric response curves for the addition of different concentrations of NaNO_2_ for the Pd-Pt/PGaN electrode in a 0.1 M PBS solution. The inset is the amplification of the current response. (**b**) and (**c**) are the corresponding calibration plots in two concentration ranges of amperometric responses vs. nitrite concentration. (**d**) Amperometric responses of Pd-Pt/PGaN sensors for the addition of 0.1 mM (a) NaNO_2_ and 5 mM common interfering substance (b-i) uric acid (UA), ascorbic acid (AA), dopamine (DA), glucose (GLU), Na_2_CO_3_, KNO_3_, GaCl_2_ and MgSO_4_.

**Figure 6 sensors-19-00606-f006:**
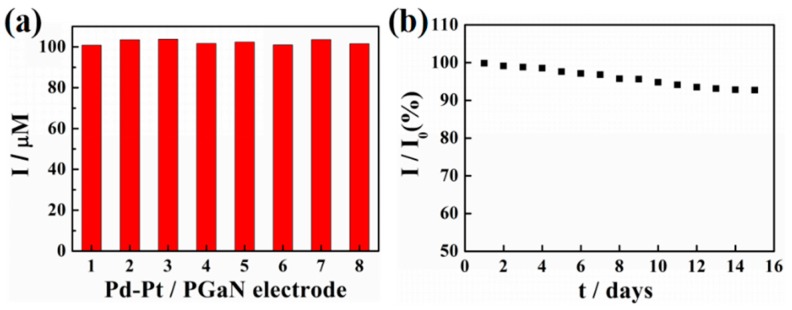
(**a**) Current responses of eight Pd-Pt/PGaN electrodes to 1 mM NaNO_2_ in 0.1 M PBS. (**b**) The long-term stability of a Pd-Pt/PGaN electrode in 15 days.

**Table 1 sensors-19-00606-t001:** Sensing performance compared to other electrode substrates for nitrite determination.

Electrode	Sensitivity (µA/mM)	Linear Range (µM)	Detection Limit (µM)	Reference
**Pt nanoparticles/GCE**	88.5	1.2–900	0.4	[33]
**Urchin-like Pd/SWCNT**	38	2–238	0.25	[34]
**Pd/graphite**	6.5	0.3–50.7	0.071	[23]
**Pt/Au**	-	10–1000	5.0	[35]
**PdFe alloy/GCE**	-	500–25500	0.8	[36]
**Hb/Au/GCE**	-	4–350	1.2	[37]
**Pd-Pt/PGaN**	150 73	1–300 300–3000	0.95	This work

**Table 2 sensors-19-00606-t002:** Determination of nitrite at various concentrations in tap and lake water.

Sample	Theoretical (µM)	Found (µM)	Recovery (%)	R.S.D (%)
	20.0	21.2	106.0	2.8
**Tap water**	40.0	38.92	97.3	3.1
	60.0	61.43	102.4	2.9
	20.0	19.52	97.6	2.6
**Lake water**	40.0	38.46	96.2	3.2
	60.0	62.63	104.4	3.5

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
