# Peer review of "Electrodeposition of Pd-Pt Nanocomposites on Porous GaN for Electrochemical Nitrite Sensing"

_sensors, 2019, doi:10.3390/s19030606_

Round 1
Reviewer 1 Report
The manuscript titled “Electrodeposition of Pd-Pt nanocomposites on porous Gallium Nitride for electrochemical nitrite sensing” describes nitrite amperometric sensor based on Pd-Pt bimetallic catalyst electrodeposited on porous GaN electrode. The presentation of the datas and styles of writing is not upto the standard of a scientific article. Starting with the abstract, it is full of grammatical and factual errors, typos, non-scientific words and incomplete sentences. Introduction is very poor and lacks a proper background, motivation and even the precise reasons to use GaN and Pt-Pd nanomaterials for nitrite sensing. The results and discussions section lacks scientific explanations of the observations. Thus overall, article misses scientific soundness, precision and was very hard and boring to read. The detail reasons have been appended below;
Every sentences of the abstract has either a typos (nitrite as nitride, μM as M) , grammatical error and confusing sentences.
In the introduction part, the authors should give a sound background of existing nitrite electrochemical sensors and should highlight how their approach is unique and can be competitive with existing nitrite sensors as well as can overcome the limitations of present nitrite sensors.
The precise reasons to use porous GaN as sensing electrode and platinum-palladium coating over it must be highlighted in the introduction part. Relevant references should be cited. For example, Electroanalysis 2005, 17, No. 5 – 6 (DOI: 10.1002/elan.200403191) gives an account of GaN to detect anionic analytes.
In Fig.1a, hexagon pore shape were attributed to the crystal structure of GaN (0001). This is not true. I recommend the author to read the article (Scientific Reports volume 7, Article number: 44063 (2017) ). To support this claim, the author must cite a reference.
Authors claim the formation of pt-pd bimetallic nanomaterial. But a proper characterizations to confirm this is missing. Even the XRD explanations presented is not precisely written. The recent article gives a nice description of the formation of the same. J. Phys. Chem. C 2012, 116, 12265−12274.
Figure 4 indicates the CVs in the presence and absence of nitrite for pt-pd modified GaN electrode and only GaN electrode. It is surprising that porous GaN without pt-pd coating is insensitive towards nitrite. Authors must highlight the reasons to use GaN as a substrate, why not cheaper alternative like carbon electrode was employed.
In the interference studies, authors claimed that sensors are not sensitive towards dopamine, ascorbic acid, uric acid and glucose. But this is not true. Pt and Pd nanomaterials are known to detect these analytes electrochemically. (a) Colloids and surfaces. B, Biointerfaces, Vol: 111, Page: 392-97 for DA, AA, UA detection using Pt-Pd nanomaterials and (b) Talanta, ISSN: 1873-3573, Vol: 99, Page: 1062-67 for glucose sensing
Beside these factual errors, article is full of grammatical errors, typos, confusing sentences and use of non-scientific English. A rigorous corrections for grammar, typos and scientific style of writing are needed.
Author Response
For referee 1:
1. Every sentences of the abstract has either a typos (nitrite as nitride, μM as M) , grammatical error and confusing sentences.
Thank you for your good suggestion, the abstract part of the manuscript has been completely revised.
"In recent years, nitrite pollution becomes a subject of great concern in human living, involving a number of fields, which include environment, food industry and biological process. However, effective detection of nitrite is an instant demanded as well as an unprecedented challenge. Here, a novel nitrite sensor was fabricated by electrochemical deposition of palladium and platinum (Pd-Pt) nanocomposites on porous gallium nitride (PGaN). The obtained Pd-Pt/PGaN sensor provides abundant electrocatalytic sites, endowing it with excellent performances for nitrite detection. The sensor also shows low detection limit of 0.95 µM, superior linear ampere response and high sensitivity (150 µA/mM for 1 to 300 µM and 73 µA/mM for 300 to 3000 µM) for nitrite. In addition, the Pd-Pt/PGaN sensor was applied and evaluated in the determination of nitrite from the real environmental samples. The experimental results demonstrate that the sensor has good reproducibility and long-term stability. It provides a practical way for rapidly and effectively monitoring nitrite content in the practical application."
2. In the introduction part, the authors should give a sound background of existing nitrite electrochemical sensors and should highlight how their approach is unique and can be competitive with existing nitrite sensors as well as can overcome the limitations of present nitrite sensors.
The introduction part of the manuscript has been improved.
More references about the nitrite electrochemical sensors were cited in the references [14-26].
The research progress and limitations on these nitrite sensors were analyzed.
"Preparation of chemically modified electrode is the focus and difficult point in detecting nitrite by electrochemical analysis. The purpose of chemically modified electrodes is to carry out the sensing material design on the electrode surface. Up to date, many researches have been reported on electrode modified materials, such as carbon material graphene [13], metal-organic frameworks (MOFs) [14] and enzyme [15]. Carbon materials have good conductivity and can be commonly used as substrate for supporting electro-active substances. Metal-organic frameworks (MOFs) possess the characteristics of high surface area and porosity. Enzyme shows highly specificity and the efficient catalysis. However, some problems existed on the above modified electrode are still not well solved in practice. Firstly, most modified materials are mainly adsorbed to the surface of the electrode by physical adsorption. In the long-term use, modified materials will peel off due to the scrapes and mechanical actions, affecting the electrode catalytic activity and lifetime. Secondly, modified electrodes are not suitable for harsh solution environments. For instance, graphene is easy to be decomposed in strong acidic solutions. The activity of enzyme can be easily affected by temperature and the pH value of solution, it is hard to guarantee the stabilization of the electrode." were added.
3. The precise reasons to use porous GaN as sensing electrode and platinum-palladium coating over it must be highlighted in the introduction part. Relevant references should be cited. For example,Electroanalysis 2005, 17, No. 5–6 (DOI: 10.1002/elan.200403191) gives an account of GaN to detect anionic analytes.
The precise reasons to use porous GaN and platinum-palladium nanoparticles have been highlighted in the introduction part. Relevant references about the anionic detection were cited.
"Thus, the properties and structures of sensing electrode are particularly important. However, gallium nitride (GaN), a representative wide bandgap semiconductor, could be a promising electrode candidate owning to its large potential window, high electron mobility and strong metal-semiconductor interaction for electrochemical sensor [16-18]. Compared with other electrode materials, GaN can achieve higher chemical and thermal stability. These unique properties make it generate less noise, resulting in a low background signal. Moreover, porous GaN (PGaN) with different pore sizes and shapes has been easily obtained by wet etching. Compared with planar GaN, PGaN provides higher surface area and more defective sites. The correlation researches indicated, it can be applied in some fields as an excellent electrode [19-21].
Due to the excellent electrocatalytic activities, noble metal nanomaterials (NMs) have been applied to composite electrochemical sensors [22-26]. In comparison with other nanomaterials, NMs are considerably more stable under ambient conditions. Moreover, their unique catalytic performances together with their decreased dimensions and high surface-to-volume ratio demonstrate enormous potentials in environmental and biological applications."were added.
4. In Fig.1a, hexagon pore shape were attributed to the crystal structure of GaN (0001). This is not true. I recommend the author to read the article (Scientific Reports volume 7, Article number: 44063 (2017) ). To support this claim, the author must cite a reference.
The pore shape is hexagonal, which is consistent with the wurtzite structure (hexagonal system) of GaN. Relevant references were cited. (Wet etching of GaN, AlN, and SiC: a review (10.1016/j.mser.2004.11.002))
"It can be seen that the pore shape is nearly hexagon, which is in accord with the hexagonal system structure of GaN." were modified.
5. Authors claim the formation of pt-pd bimetallic nanomaterial. But a proper characterizations to confirm this is missing. Even the XRD explanations presented is not precisely written. The recent article gives a nice description of the formation of the same. J. Phys. Chem. C 2012, 116, 12265−12274.
Pd and Pt (Pd-Pt) nanocomposites were preparated by a two-step electrodeposition method.
The Pd-Pt nanoparticle is a composite material, not a core-shell configuration. Experiment evidences were consistent with literature reports (Analytical Letters, 2016: 00032719. 2016.1159694, Journal of Power Sources, 2010, 195(2):445-452, Electrochemistry Communications, 2012, 20(20):56-59.) The XRD explanations were modified.
"Compared with the diffraction peak of Pt (111) crystal face at 39.7°, the Pd-Pt nanocomposites peaks are shifted to higher angle due to the combination of Pd to Pt. The results of experiment correspond with literature reports [29,30]. No characteristic peaks of Pd or Pt oxides were detected. Therefore, the nanocomposites composed of Pt and Pd were successfully deposited on the PGaN by electrochemical method." were added.
6. Figure 4 indicates the CVs in the presence and absence of nitrite for pt-pd modified GaN electrode and only GaN electrode. It is surprising that porous GaN without pt-pd coating is insensitive towards nitrite. Authors must highlight the reasons to use GaN as a substrate, why not cheaper alternative like carbon electrode was employed.
Porous GaN without Pd-Pt coating is insensitive towards nitrite, That is, no distinct current response of nitrite is observed at the PGaN electrode. Thus, the bare PGaN electrode provides a low background signal.
"Meanwhile, no distinct current response was observed at the PGaN (curve b1 and b2). That is, the bare PGaN electrode provides a low background signal, which is beneficial for sensor to generate less noise and recognize weak signals. To further illustrate the excellent catalytic performance of Pd-Pt/PGaN, the effects of Pt/PGaN and Pd/PGaN on direct oxidation of nitrite ion were studied by CV. As shown in Figure 4b, the Pt/PGaN electrode demonstrated a small current response to 1 mM nitrite, while no significant current change was observed on the Pd/PGaN electrode. The results show that Pd-Pt bimetallic complex has higher catalytic activity than Pt single metal in detection of nitrite." were added. Relevant experiments were supplemented. Figure 4. (b) CVs of the Pt/PGaN, Pd/PGaN and Pd-Pt/PGaN electrodes in 0.1 M PBS buffer containing 1 mM NaNO2.
7. In the interference studies, authors claimed that sensors are not sensitive towards dopamine, ascorbic acid, uric acid and glucose. But this is not true. Pt and Pd nanomaterials are known to detect these analytes electrochemically. (a) Colloids and surfaces. B, Biointerfaces, Vol: 111, Page: 392-97 for DA, AA, UA detection using Pt-Pd nanomaterials and (b) Talanta, ISSN: 1873-3573, Vol: 99, Page: 1062-67 for glucose sensing
GaN supported noble metal nanoparticles were used in the field of electrochemical detection. Our previous studies had demonstrated GaN-based electrodes could be a promising support for metal nanoparticles due to their good electric conductivity and strong interaction between metal and GaN surface. Owing to the existence of GaN, the metal-GaN electrode has good selectivity for detecting substances. Relevant references were cited, such as (a) Sensors & Actuators B Chemical, 2017, 240:142-147. (b) Talanta, 2017:S0039914017304988.S. (c) J. Electrochem. Soc. 162 (2015) D625–D629. (d) Electrochim. Acta 130 (2014) 537–542.
In these interference studies, the current responses were hardly observed when DA, AA, UA and GLU were added respectively. The present result is good agreement with the reported experiments.
8. Beside these factual errors, article is full of grammatical errors, typos, confusing sentences and use of non-scientific English. A rigorous corrections for grammar, typos and scientific style of writing are needed.
Grammar, typos and scientific style of writing have been rigorously revised.

Reviewer 2 Report
This paper describes amperometric determination of nitrite ion using Pd-Pt/PGaN electrode. The manuscript contains fabrication processes of Pd-Pt/PGaN electrode, electrochemical characteristics of Pd-Pt/PGaN electrode and analytical applications for nitrite ion. However, additional experimental results are needed and some descriptions and explanations are not clear. Therefore, the points listed below to be explained and modified before publication with major revisions.
Describe more recent progress for electrochemical determination of nitrite ion and compare the need and differentiation of this paper in the introduction section.
Show and explain whether platinum or palladium affect nitrite ion detection by cyclic voltammetry
A nitrite ion acts as weak base and the composition of nitrite ion and nitrous acid changes with pHs. For best determination of nitrite ion, the effect of the pHs on the of nitrite determination should be studied.
Describe the shape and dimension of working electrode.
Amperometric analysis influence by the applied potential and it should be determined by experimental result. Describe the effect of applied potential for nitrite ion determination.
Explain the process of obtaining the detection limit.
The interference of small organic molecules of UA, AA, DA were studied. Show the interference of cations and inorganic anions in body or soil.
Abstract; 0.95 Mà0.95 μM, 300 Mà300 μM, 3000 Mà3000 μM
Correct grammatical errors and style of sentence.

Author Response
For referee 2:
1. Describe more recent progress for electrochemical determination of nitrite ion and compare the need and differentiation of this paper in the introduction section.
Thank you for your good suggestion. In the introduction part, the research progress for electrochemical determination of nitrite was added, the need and differentiation of Pd-Pt/PGaN sensor were analysed. Relevant references were cited.
"Preparation of chemically modified electrode is the focus and difficult point in detecting nitrite by electrochemical analysis. The purpose of chemically modified electrodes is to carry out the sensing material design on the electrode surface. Up to date, many researches have been reported on electrode modified materials, such as carbon material graphene [13], metal-organic frameworks (MOFs) [14] and enzyme [15]. Carbon materials have good conductivity and can be commonly used as substrate for supporting electro-active substances. Metal-organic frameworks (MOFs) possess the characteristics of high surface area and porosity. Enzyme shows highly specificity and the efficient catalysis. However, some problems existed on the above modified electrode are still not well solved in practice. Firstly, most modified materials are mainly adsorbed to the surface of the electrode by physical adsorption. In the long-term use, modified materials will peel off due to the scrapes and mechanical actions, affecting the electrode catalytic activity and lifetime. Secondly, modified electrodes are not suitable for harsh solution environments. For instance, graphene is easy to be decomposed in strong acidic solutions. The activity of enzyme can be easily affected by temperature and the pH value of solution, it is hard to guarantee the stabilization of the electrode."
"Thus, the properties and structures of sensing electrode are particularly important. However, gallium nitride (GaN), a representative wide bandgap semiconductor, could be a promising electrode candidate owning to its large potential window, high electron mobility and strong metal-semiconductor interaction for electrochemical sensor [16-18]. Compared with other electrode materials, GaN can achieve higher chemical and thermal stability. These unique properties make it generate less noise, resulting in a low background signal. Moreover, porous GaN (PGaN) with different pore sizes and shapes has been easily obtained by wet etching. Compared with planar GaN, PGaN provides higher surface area and more defective sites. The correlation researches indicated, it can be applied in some fields as an excellent electrode [19-21]."
"Due to the excellent electrocatalytic activities, noble metal nanomaterials (NMs) have been applied to composite electrochemical sensors [22-26]. In comparison with other nanomaterials, NMs are considerably more stable under ambient conditions. Moreover, their unique catalytic performances together with their decreased dimensions and high surface-to-volume ratio demonstrate enormous potentials in environmental and biological applications. "were added.
2. Show and explain whether platinum or palladium affect nitrite ion detection by cyclic voltammetry.
Relevant experimental data was supplemented.
Figure 4. (b) CVs of the Pt/PGaN, Pd/PGaN and Pd-Pt/PGaN electrodes in 0.1 M PBS buffer containing 1 mM NaNO2.
"To further illustrate the excellent catalytic performance of Pd-Pt/PGaN, the effects of Pt/PGaN and Pd/PGaN on direct oxidation of nitrite ion were studied by CV. As shown in Figure 4b, the Pt/PGaN electrode demonstrated a small current response to 1 mM nitrite, while no significant current change was observed on the Pd/PGaN electrode. The results show that Pd-Pt bimetallic complex has higher catalytic activity than Pt single metal in detection of nitrite." were added.
3. A nitrite ion acts as weak base and the composition of nitrite ion and nitrous acid changes with pHs. For best determination of nitrite ion, the effect of the pHs on the of nitrite determination should be studied.
An experimental investigation about the effect of the pHs and the conclusion in the experimental data was analysed. As show in Figure 4. (d) CV peak current values of the Pd-Pt/PGaN electrode in 0.1 M PBS buffer containing 1 mM NaNO2 under different pH conditions (2.0, 3.0, 5.0, 7.0, 9.0, 11.0 and 12.0).
"In addition, the effects of Pd-Pt/PGaN electrode with different pH conditions on the detection of nitrite were studied in Figure 4d. The results show that the peak current values of electrode increase as pH of the solution increase in the range of 2.0 to 7.0, and then begin to fall rapidly. Thus, 7.0 was selected as the optimum pH condition." were added.
4. Describe the shape and dimension of working electrode.
In the experiment part, "The size of GaN chips was 0.3 cm×1.5 cm." were added.
5. Amperometric analysis influence by the applied potential and it should be determined by experimental result. Describe the effect of applied potential for nitrite ion determination.
Amperometric analysis influence by the applied potential and it was determined in Figure 4. (a). A conspicuous oxidation peak of the Pd-Pt/PGaN electrode was observed about 100 µA at 0.65 V.
"Figure 4a shows the electrocatalytic responses of the Pd-Pt/PGaN and PGaN electrodes to 1 mM NaNO2 at the potential range from 0.2 to 1.2 V. Compared with the CV curve a2, a conspicuous oxidation peak of the Pd-Pt/PGaN electrode was observed about 100 µA at 0.65 V in curve a1."
6. Explain the process of obtaining the detection limit.
The formula of the detection limit is LOD=3Sb/b, where Sb denotes the standard deviation of current in blank solution and b denotes the slope of the fitting curve. In this experiment, phosphate buffer solution (pH=7) was served as blank control. The standard deviation of current was 4.75×10-8 A. By calculation, the detection limit was 0.95 µM.
7. The interference of small organic molecules of UA, AA, DA were studied. Show the interference of cations and inorganic anions in body or soil.
Amperometric responses of Pd-Pt/PGaN sensor for the addition of 0.1 mM (a) NaNO2 and 5 mM common interfering substance (b-i) uric acid (UA), ascorbic acid (AA), dopamine (DA), glucose (GLU), Na2CO3, KNO3, GaCl2, and MgSO4 was added in Figure. 2d.
"When 5 mM UA, AA, DA, GLU, Na2CO3, KNO3, GaCl2 and MgSO4 were added into PBS solution respectively, no amperometric responses were detected." were added.
8. Abstract; 0.95 Mà0.95 μM, 300 Mà300 μM, 3000 Mà3000 μM
"The nitrite sensor shows low detection limit of 0.95 µM, superior linear ampere response and high sensitivity (150 µA/mM for 1 to 300 µM and 73 µA/mM for 300 to 3000 µM) for nitrite "were modified.
9. Correct grammatical errors and style of sentence.
Major revisions have been made to the full text.

Reviewer 3 Report
In this work Xi and coworkers demonstrate a novel approach to the electrochemical detection of nitrites. The whole structure of the manuscript looks solid and convincing. The reader clearly understands motivations, targets and methodology in the introduction. In the results and discussion section of the manuscript, the authors report the realization and structural as well as morphological characterization of the sensors. Then, authors focus on the electrochemical, electrocatalytic and amperometric response of their samples. Finally, the full characterization of the sensors in different operation regimes and against different sources of nitrite is reported and discussed.
Regarding the presentation and discussion of the results, some improvements should be implemented.
Page 1 row 33: Electrochemistry is not a technique
In general, the use of English language within the manuscript should be improved.
GaN and P-GaN: An alternative strategy to achieve porous systems of GaN (or other III-V semiconductors) could be resorting to assemblies of nanowires grown using top-down or bottom-up approaches. Nanowires systems display in fact high surface to volume ratio and are somehow more controllable respect to porous systems obtained by wet etching, which make them promising for sensing applications. See for instance the reference F. Floris et al., Nanomaterials 2017, 7(11), 400; https://doi.org/10.3390/nano7110400, Self-Assembled InAs Nanowires as Optical Reflectors. Could the authors briefly comment on this point, and/or on other alternative strategies to the realization of the electrodes?
I found a bit misleading the sequence of figures 2-3-4: Fig 2 d shows electrical data while panels a-c display the morphology of the samples; Fig 3 reports info about the structure and chemistry; Fig 4 reports electrical data. I would suggest to make the figures more homogeneous, for instance: put the results of morphological (fig2-a-b-c) and structural (fig.3) characterization together in new fig 2; put the results of electrochemical measurements (panel 2.d and fig 4) all together in new figure 3.
Author Response
For referee 3:
1. Regarding the presentation and discussion of the results, some improvements should be implemented.
Thank you for your good suggestion. Major revisions have been made to the full text.
2. Page 1 row 33: Electrochemistry is not a technique.
At present, various techniques are used to detect nitrite, such as fluorescence spectrometry [7], chromatography [8], chemiluminescence [9] and electrochemical method [10].
3. In general, the use of English language within the manuscript should be improved.
The manuscript has been completely revised.
4. GaN and P-GaN: An alternative strategy to achieve porous systems of GaN (or other III-V semiconductors) could be resorting to assemblies of nanowires grown using top-down or bottom-up approaches. Nanowires systems display in fact high surface to volume ratio and are somehow more controllable respect to porous systems obtained by wet etching, which make them promising for sensing applications. See for instance the reference F. Floris et al., Nanomaterials 2017, 7(11), 400; https://doi.org/10.3390/nano7110400, Self-Assembled InAs Nanowires as Optical Reflectors. Could the authors briefly comment on this point, and/or on other alternative strategies to the realization of the electrodes?
Based on our experimental experience, the wet etching process of native GaN and P-GaN to achieve porous structure is not easy compared with n-GaN. The reported typical etching morphologies are mainly focused on the pore structure, yet GaN NWs have rarely been reported concerning the detailed etching process. The use of sensors depends on the structure of sensing materials. Different structures can exhibit its advantages for different fields. For instance, the native GaN electrode showed no response to cations, but could be used in cation sensing after a cation selective membrane was deposited. InN thin films were used in a similar fashion for anion detection, taking advantage of their high surface charge densities and positively charged surface donor states(Appl. Phys. Lett., 2007, 91, 202109). The voltammetric detection was implemented on surfacefunctionalized GaN nanowires electrodes for DNA hybridization(J. Mater. Chem., 2009, 19, 928-933). Based on the sensing mechanisms, the nanowires systems is more suitable for optical sensing via PL and Raman spectroscopy(Nano Lett., 2012, 12, 6180–6186.).
5. I found a bit misleading the sequence of figures 2-3-4: Fig 2 d shows electrical data while panels a-c display the morphology of the samples; Fig 3 reports info about the structure and chemistry; Fig 4 reports electrical data. I would suggest to make the figures more homogeneous, for instance: put the results of morphological (fig2-a-b-c) and structural (fig.3) characterization together in new fig 2; put the results of electrochemical measurements (panel 2.d and fig 4) all together in new figure 3.
According to the advice, we made adjustments on the order of images.
Figure 2. SEM images of (a) Pt/PGaN, (b) Pd/PGaN and (c) Pd-Pt/PGaN electrodes.
Figure 4. (a) CVs of Pd-Pt/PGaN electrode (a1 and a2) and PGaN electrode (b1 and b2) in 0.1 M PBS (pH 7) solution in the presence (a1 and b1) and absence (a2 and b2) containing 1 mM NaNO2. (b) CVs of the Pt/PGaN, Pd/PGaN and Pd-Pt/PGaN electrodes in 0.1 M PBS buffer containing 1 mM NaNO2. (c) Electrochemical impedance spectroscopy (EIS) spectra of three electrodes in 0.1 M KCl containing 5 mM K3[Fe(CN)6]/K4[Fe(CN)6]. The inset is Randles equivalence circuit model. (d) CV peak current values of the Pd-Pt/PGaN electrode in 0.1 M PBS buffer containing 1 mM NaNO2 under different pH conditions (2.0, 3.0, 5.0, 7.0, 9.0, 11.0 and 12.0).

Round 2
Reviewer 1 Report
The modified version of the article "Electrodeposition of Pd-Pt nanocomposites on porous GaN for electrochemical nitrite sensing" is much better. Authors have properly answered the questions raised as well as have performed changes in the article.
I have a few minor points to comments regarding Grammar and typos;
1. Some grammatical errors in the abstract like "In recent years, nitrite pollution becomes a subject...." should be "In recent years, nitrite pollution has become a subject...."
In 3rd line of abstract, it should be "demand" not demanded.
2. In 44th line " Enzyme shows highly specificity...." should be "Enzyme shows high specificity..."
3. In 45th line " some problems existed on the above modified electrode are..." should be "some problems existed on the above modified electrodes are..."
4. In 47th line, there is a typo "scrapes". It should be "scratches"
5. In 59th line, defective sites should be defects
6. In 108th line "which is in accord with the hexagonal...." should be "which is in accordance with the hexagonal..."
7. In 178th line RCT should be Rct
8. In 225th line "On the basic of its high...." should be "On the basis of its high....
Author Response
For referee 1:
1. Some grammatical errors in the abstract like "In recent years, nitrite pollution becomes a subject...." should be "In recent years, nitrite pollution has become a subject...."
In 3rd line of abstract, it should be "demand" not demanded.
Thank you for your good suggestion, grammatical errors in the abstract have been corrected.
"In recent years, nitrite pollution has become a subject of great concern in human living, involving a number of fields, which include environment, food industry and biological process. However, effective detection of nitrite is an instant demand as well as an unprecedented challenge."
2. In 44th line " Enzyme shows highly specificity...." should be "Enzyme shows high specificity..."
According to your advice, the sentence has been revised.
"Enzyme shows high specificity and the efficient catalysis."
3. In 45th line " some problems existed on the above modified electrode are..." should be "some problems existed on the above modified electrodes are..."
Errors in sentence have been corrected.
"However, some problems existed on the above modified electrodes are still not well solved in practice."
4. In 47th line, there is a typo "scrapes". It should be "scratches"
A typo in sentence has been corrected.
"In the long-term use, modified materials will peel off due to the scratches and mechanical actions, affecting the electrode catalytic activity and lifetime."
5. In 59th line, defective sites should be defects.
The word has been revised.
"Compared with planar GaN, PGaN provides higher surface area and more defects."
6. In 108th line "which is in accord with the hexagonal...." should be "which is in accordance with the hexagonal..."
The sentence has been revised.
"It can be seen that the pore shape is nearly hexagon, which is in accordance with the hexagonal system structure of GaN."
7. In 178th line RCT should be Rct.
The word has been revised.
"And the increase of surface active region also leads to the decrease of Rct."
8. In 225th line "On the basic of its high...." should be "On the basis of its high....
The word has been revised.
"On the basis of its high electron mobility and special surface structure."

Reviewer 2 Report
The following sentences in Introduction section were unnecessary or improper expression and need to correct it.
“Preparation of chemically modified electrode is the focus and difficult point in detecting nitrite by electrochemical analysis.”
Author Response
For referee 2:
1. The following sentences in Introduction section were unnecessary or improper expression and need to correct it. “Preparation of chemically modified electrode is the focus and difficult point in detecting nitrite by electrochemical analysis.”
The sentence has been deleted.
